# Optimization of spatial distribution of sports parks based on accessibility analysis

**Kairan Yang**[1,2☉], **Yujun Xie**[3☉], **Hengtao Guo** [ID][1,2]*

**1** College of Physical Education, Hunan Normal University, Changsha, Hunan, China, **2** Key Laboratory of Physical Fitness and Sports Rehabilitation of Hunan Province, Changsha, Hunan, China, **3** Changzhou Institute of Building Science, Changzhou, Jiangsu, China

☉ These authors contributed equally to this work.
* 76926280@qq.com

## Abstract

In recent years, public sports services have attracted great attention owing to their increasingly important role in public health. However, effective evaluation metrics measuring the efficiency of such services from a spatial perspective (e.g., accessibility and distribution of sports parks) remain absent. Indeed, most designs of sports park distribution in urban areas did not consider practical factors such as local road networks, population distribution, and resident preference, resulting in low utilization rates of these parks. In this study, a spatial accessibility-based method is proposed for evaluation of the distributions of sports parks. As a demonstration, the distribution of sports parks in the central urban area of Changsha, China was investigated using the proposed method by the GIS network analysis. Additionally, optimization strategies for sports park distribution (in terms of spatial distribution and overall accessibility) were developed by using spatial syntax.

## 1 Introduction

As is known, sports can directly improve physical and mental health of human being. In 2014, "fitness for all" was officially proposed as a national strategy of China and sports and leisure activities have becoming a key part of daily life of urban citizens since then [1]. The *Healthy China 2030 Planning Outline* issued in 2016 emphasized the significance of public fitness facilities and sports parks, and officially launched a plan for the construction of new sports parks all over China [2]. A sports park refers to a specialized park with complete sports and fitness facilities for competitions and training of athletes and daily sports and leisure activities of urban citizens. As an integration of sports and leisure facilities and ecological gardens, sports parks have been playing a key role in public health and the national strategy of Healthy China by providing sports avenues to urban residents, thereby enhancing their quality of life [3,4]. However, existing sports parks in China exhibit poor spatial distribution and low accessibility owing to absences of studies and guidance in this field, resulting in low utilization rate [5,6]. In recent years, an increasing population of urban residents have been concerned about accessibility to sports parks and services, namely sharing of eco-sports resource by the public. Currently, evaluations of sports parks were achieved based on sports park area per capita, facility

**Data Availability Statement:** All relevant data are within the paper and its Supporting Information files.

**Funding:** This research was funded by the National Key Research and Development Plan project (approval number: 2022YFC2010203), which was

presided over by Zheng LAN, College of Physical Education, Hunan Normal University.

**Competing interests:** The authors have declared that no competing interests exist.

improvement, and service satisfaction, while other key factors such as accessibility by residents and spatial distribution are typically not taken into consideration [7,8].

The accessibility is defined as the size of the interaction between nodes within the road transportation system, or the ease or difficulty of moving from one place to another [9]. Likewise, the accessibility of sports parks is defined as the ease or difficulty for urban residents to participate in sports activities at nearby sports parks. It is influenced by both objective (e.g., geological distance, road network, and transportation facilities) and subjective factors (attitudes and preferences of the residents) [10]. To date, various studies of sports park accessibility have been reported. For instance, Zhong *et al.* analyzed the shortcomings of the current policies on sports parks in China on the basis of the policy models in Japan and proposed several solutions to policy optimization, including control of construction scale, refining of licensing procedures, promotion of social participation, and development of relevant standards [11]. Stanley *et al.* explored potential driving factors of high-quality development of sports parks and proposed strategies such as innovative land use and management, coordinated internal/external development, propaganda, involvement of cultural diversity, and improvement of security services [12]. Sun *et al.* developed an innovative city model by integrating artificial intelligence and architecture [13]. However, these studies focused on theoretical aspects (e.g., concept, measurement methods, and influencing factors), while applications of sports park accessibility in distribution optimization of sports venues, especially sports parks, have been rarely reported. Indeed, most designs of sports park distribution in urban areas did not consider practical factors such as local road networks, population distribution, and resident preference, resulting in low utilization rates of these parks [14]. Hence, the evaluation systems of sports parks shall be improved. A possible solution is spatial syntax involving rational accessibility indicators. In this case, it is possible to meet daily fitness needs of urban residents and this method may serve as a key part of the national strategy of Healthy China [15]. Spatial syntax refers to an analysis tool for spatial configurations and human activity patterns in buildings and urban areas. It addresses where people are, how they move, how they adapt, how they develop and how they talk about it [16].

In this study, a spatial accessibility-based method is proposed for evaluation of the distributions of sports parks. As a demonstration, the distribution of sports parks in the central urban area of Changsha, China was investigated using the proposed method by the GIS network analysis. Additionally, optimization strategies for sports park distribution (in terms of spatial distribution and overall accessibility) were developed by using spatial syntax. This study provides references and guidance to the design of sports park distribution.

## 2 Methods

### 2.1 Spatial accessibility-based analysis of sports parks

To date, the spatial distributions of sports parks in China, as well as the corresponding influencing factors, have been investigated by using different methods (Table 1), including grid dimension analysis, spatial autocorrelation analysis, and kernel density estimation [17,18]. Indeed, the service capability of sports parks is closely related to their accessibility in a positive manner [19]. Theoretically, a short straight-line distance from residential to sports parks results in a short journey and high accessibility. Due to the complex urban road network and abundant road barriers, however, it is necessary to consider other factors besides straight-line distance, including spatial distribution, urban road network, and transportation facilities [20]. Since the road data was retrieved from the OpenStreetMap (OSM) database, the network analysis method was employed in this study in virtue of its high effectiveness and accuracy.

**Table 1. Features of different accessibility analysis methods.**

| Measurement Method | Quantitative Simplicity | Data Accessibility | Evaluation Subjectivity | Spatial Layout | Reachability Accuracy |
|---|---|---|---|---|---|
| Buffer Analysis | Simple | Simple | Somewhat Subjective | Lacking | Inaccurate |
| Statistical Indicators | Simple | Simple | Somewhat Subjective | Lacking | Inaccurate |
| Minimum Proximity | Moderate | Moderate | Somewhat Subjective | Lacking | Relatively Accurate |
| Cost Weighting | Moderate | Moderate | Somewhat Subjective | Present | Relatively Accurate |
| Gravity Potential | Difficult | Difficult | Subjective | Lacking | Relatively Accurate |
| Two-step Search | Difficult | Difficult | Objective | Present | Accurate |
| Network Analysis | Moderate | Difficult | Objective | Present | Accurate |

## 2.2 Overview of the study area

Changsha is the capital of Hunan Province, China and has a population of 10.4 million by 2022 [21]. This study focuses on the most populous central urban areas of Changsha (the Yuelu District, the Yuhua District, the Kaifu District, the Furong District, and the Tianxin District) (Fig 1). By 2021, the central urban area of Changsha has a size of 1198.2 km$^2$ and a built-up area of 357.9 km$^2$ [22]. The current area of sports parks in China is 1.715 km$^2$. A total of 13 sports parks have been constructed in the central urban area of Changsha, with four in the Kaifu District, three in the Yuelu District, three in the Yuhua District, two in the Furong District, and one in the Tianxin District. These sports parks were involved as the subjects in this study.

## 2.3 Analysis methods

**2.3.1 Network analysis.** Network analysis is an effective modelling tool for geographic networks and infrastructures. The spatial and attribute data of network elements have been explored and the network performances have been thoroughly investigated to identify the shortest path and optimize resource allocation [23]. In this way, the accessibility of sports parks can be effectively evaluated, the urban road networks can be effectively modeled, and GIS network datasets can be effectively established [24].

In this study, field research and interviews were executed to explore the main behavioral characteristics of residents in the central urban area of Changsha when using sports parks. Herein, 500 respondents were enrolled in a questionnaire (see Supporting Information) survey and the results are summarized in Table 2. As indicated, over 95% of respondents preferred a travel time to sports parks less than 15 min and over 95% of respondents preferred walking or biking as the means of transportation to sports parks. Therefore, the accessibilities of sports parks were evaluated in terms of walking and biking in this study. The accessibilities of sports parks in the central urban area of Changsha were divided into different levels shown in Table 3 [25].

The accessibilities of sports parks in the central urban area of Changsha were evaluated on the basis of the ratio of area with walking-accessibility to the total area of the sports park, the ratio of area with biking-accessibility to the total area of the sports park, the ratio of population with walking-accessibility to sports parks, and the ratio of population with biking-accessibility to sports parks [26], which can be calculated by Eqs (1)–(4), respectively:

$$\text{Ratio of area with walking} - \text{accessibility}$$
$$= (\text{area of a region where residents can walk to a sport park/total area of the study area})$$
$$\times 100\% \tag{1}$$

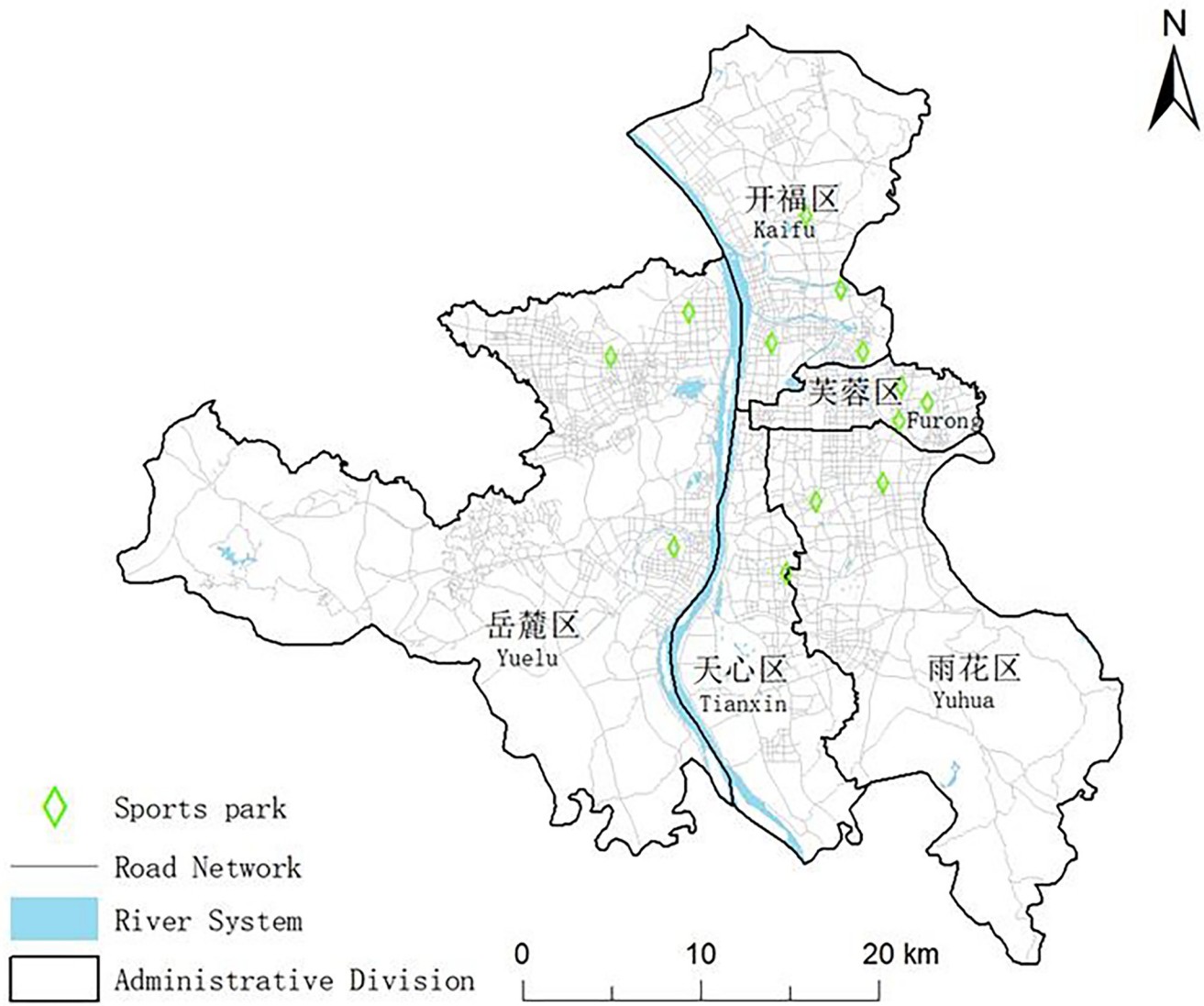

**Fig 1. Spatial distribution of sports parks in the study area (Reprinted from the Standard Map Service (http://bzdt.ch.mnr.gov.cn/) under a CC BY license, with permission from the Ministry of Natural Resources of the People's Republic of China, original copyright 2023).**

**Table 2. Preferred travel time and means of transportation to sports parks.**

| Preferred travel time (min) | Number of respondents | Preferred means of transportation | Number of respondents |
|---|---|---|---|
| 0–5 | 156 (31.2%) | Walking | 267 (53.4%) |
| 5–10 | 125 (25.0%) | Biking | 215 (43.0%) |
| 10–15 | 198 (39.6%) | Driving | 18 (3.6%) |
| 15–20 | 14 (2.8%) | | |
| > 20 | 7 (1.4%) | | |

**Table 3. Levels of sports park accessibility.**

| Travel time (min) | 0–5 | 5–10 | 10–15 |
|---|---|---|---|
| Level | one | two | three |

Ratio of area with biking−accessibility

 = (area of a region where residents can ride to a sport park/total area of the study area)

 × 100% (2)

Ratio of population with walking−accessibility to a sports park

 = (population within area with walking−accessibility/total population in the study area)

 × 100% (3)

Ratio of population with biking−accessibility to a sports park

 = (population within the area with biking−accessibility/total population in the study area)

 × 100% (4)

**2.3.2 Space syntax.** Network analysis can effectively evaluate the overall accessibility of sports parks. It has been demonstrated that a comprehensive assessment in terms of both topological structures and functions of the transportation network is essential for the development of specific optimization strategies for the accessibility of sports parks [27]. Meanwhile, space syntax provides multiple measurement indicators for analysis of the morphological structure of transportation networks, including connectivity, depth, and integration [28,29]. In this study, network analysis and space syntax were combined to clarify the topological structures and functions of transportation networks and develop an optimization strategy for the spatial distribution of sports parks.

(1) Connectivity

Connectivity ($k_i$) refers to the number of other spatial elements that interact with the $i$th spatial element. The connectivity is proportional to the spatial permeability.

(2) Mean depth

Mean depth refers to the average distance of a node to all other nodes in a network. In a road network, the mean depth reflects the convenience of the location of a specific node. Mean depth can be calculated by:

$$MD_i = \frac{\sum_{j=1}^{n} d_{ij}}{n-1} \tag{5}$$

where $d_{ij}$ refers to the shortest distance between any two points in the network, $\sum_{j=1}^{n} d_{ij}$ refers to the total depth within the region, and $n$ refers to the total number of nodes within the region.

(3) Integration

Integration comprises global integration and local integration. Global integration refers to the topological correlation of an axis with all other axes, while local integration refers to the correlation of an axis with other axes a few (typically three or five) steps away from this axis. Integration can characterize the correlation of a given space and local or overall space. It is defined as:

$$RA_i = \frac{2(MD_i - 1)}{n-2} \tag{6}$$

$$D_n = \frac{2\{n[\log_2(\frac{n+2}{3}) - 1] + 1\}}{(n-1)(n-2)} \qquad (7)$$

$$RRA_i = \frac{RA_i}{D_n} \qquad (8)$$

$$I = \frac{1}{RRA_i} \qquad (9)$$

where $n$ refers to the total number of axes in the axial network; $RA_i$ refers to the integration; $MD_i$ refers to the mean depth. As the number of nodes has a significant impact on the depth, $D_n$ is employed to normalize the integration value to eliminate its effect on integration; $RRA_i$ refers to the normalized integration. Herein, the reciprocal of the integration ($I$) is used as the integration indicator for accessibility evaluation. $I$ is proportional to the accessibility.

## 2.4 Data sources

**2.4.1 Geographic data.** The geographic data used in this study include road data and distribution of sports parks in the central urban area of Changsha. The road data was obtained from the OSM [30], remote sensing maps issued by the Landsat thematic mapper (TM), and maps issued by the General Electric (GE) Satellite. Subsequently, these sources were compared and interpreted along with the land-use classification maps in Changsha's general plan, with correction based on field survey. The distribution of sports parks was obtained by field survey. All data were accessed in 2023.

**2.4.2 Population.** The population data used in this study were extracted from the *Statistical Yearbook of Changsha* and the information provided on the official website of Changsha government. The populations of different residential communities in the central urban area of Changsha were obtained from the statistical data from the official website of Changsha government. After removing incomplete data accordingly, a total of 4303 communities were involved in this study and the population of each community was calculated based on the household information and the average population of family households. In this way, the spatial distribution of population in the central urban area of Changsha can be obtained.

## 3 Results and discussion

### 3.1 Walking- and biking-accessible area of existing sports parks

Based on the data of road network, communities, and population data in the study area, a road accessibility model was established by using ArcGIS software. The area with reasonable accessibility to existing sports parks and the population with accessibility to these parks were determined by using the network analysis method. Fig 2 shows the distribution of areas with (a) walking-accessibility and (b) biking-accessibility to sports parks in the study area. The areas with walking- and biking-accessible to sports parks in the study area are summarized in Table 4. As indicated, the cumulative walking- and biking-accessible areas of existing sports parks in the study area are 24.84 and 223.86 km$^2$, respectively, accounting for 2.08% and 18.68% of the total area, respectively. In other words, approximately 98% and 82% of the administrative area in the study area are not walking- and biking-accessible to sports parks, respectively, indicating relatively low overall accessibility of existing sports parks in the study area. Additionally, the Level 1 (0–5 min, a user-friendly level) walking- and biking-accessible

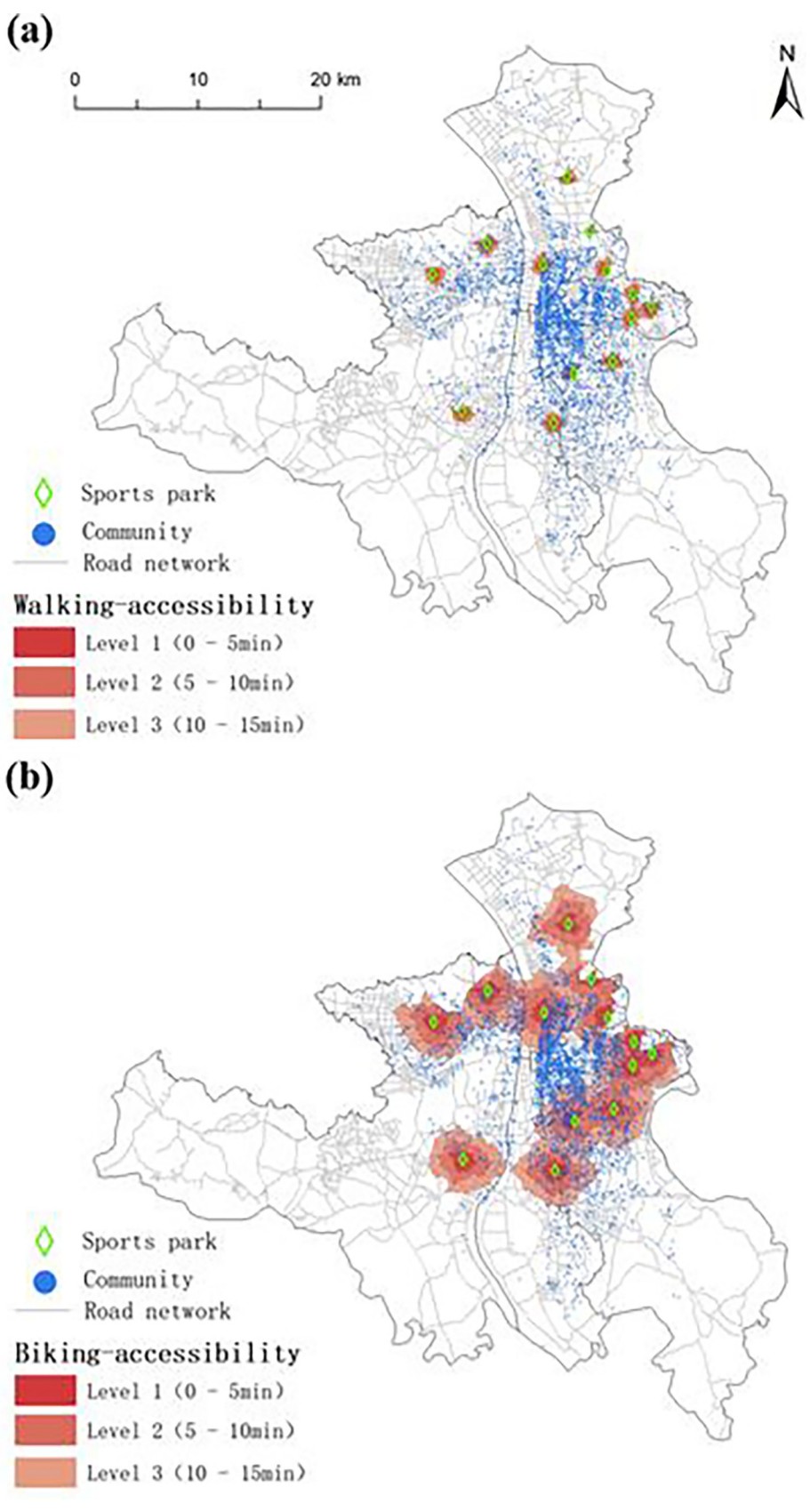

**Fig 2.** Distribution of areas with (a) walking-accessibility and (b) biking-accessibility to sports parks in the study area (Reprinted from Standard Map Service (http://bzdt.ch.mnr.gov.cn/) under a CC BY license, with permission from the Ministry of Natural Resources of the People's Republic of China, original copyright 2023).

areas in the study area are 2.59 and 30.53 km$^2$, respectively, accounting for 0.22% and 2.55% of the total area of the study area, respectively, suggesting that the service area of sports parks in the study area is extremely limited. Therefore, the accessibility of sports parks and relevant services in the study area shall be urgently improved.

The growth rate of the biking-accessible area from Level 2 to Level 3 (by 1.6%) is significantly lower than that from Level 1 to Level 2 (by 4.5%), indicating that some sports parks are concentrated and their service areas may overlap. In the optimization process, further efforts should be made to reduce the travel time for residents to reach sports parks.

The areas of Levels 1, 2 and 3 biking-accessible areas are significantly larger than those of walking-accessible areas, indicating that the road network in the study area is well-developed and effective. Nevertheless, the area ratio of biking-accessible area/walking-accessible area decreases as the accessibility level increases, suggesting that the advantage of biking over walking is dependent on the road conditions (long travel distance corresponds to increased complexity of road network).

## 3.2 Population with accessibility to existing sports parks

As shown in Fig 2, the population distribution in the study area is uneven. Thus, the area with accessibility to sports parks alone is not sufficient to evaluate the services provided by sports parks. Hence, the population with accessibility to existing sports parks in the study area is also taken into consideration.

Table 5 summarizes the population with accessibility to existing sports parks in the study area. As demonstrated, 438,900 residents (5.59% of the total residential population) have walking-accessibility to sports parks, and 4,084,500 residents (58.85% of the total residential population) have biking-accessibility to sports parks. With regard to the walking-accessibility, only 0.93% of the total residential population have Level 1 walking-accessibility to sports parks, and less than 6% of the total residential population have walking-accessibility to sports parks, suggesting that the walking-accessibility in the study area is low. With regard to the biking-accessibility, 7% of the total residential population have Level 1 biking-accessibility to sports parks, and overall half (58.85%) of the total residential population have biking-accessibility to sports parks, both of which are significantly higher than those of walking-accessibility to sports parks.

In summary, both the ratio of area with accessibility to sports parks and the ratio of population with accessibility to sports parks can reflect the accessibility of sports parks in the study area. However, uneven population distribution in the study area makes it more reasonable to evaluate the accessibility of sports parks by using the ratio of population with accessibility to

**Table 4. Areas with walking- and Biking-accessibility to existing sports parks in the study area.**

| Accessibility level | Walking-accessibility | | Biking-accessibility | |
|---|---|---|---|---|
| | Areas with accessibility /km$^2$ | Percentage/% | Areas with accessibility /km$^2$ | Percentage/% |
| Level 1 | 2.59 | 0.22 | 30.53 | 2.55 |
| Level 2 | 8.28 | 0.69 | 87.01 | 7.26 |
| Level 3 | 13.97 | 1.17 | 106.32 | 8.87 |
| Total | 24.84 | 2.08 | 223.86 | 18.68 |

**Table 5. Population served by existing sports parks in the study area.**

| Accessibility level | Walking-accessibility | | Biking-accessibility | |
|---|---|---|---|---|
| | Population served | Percentage/% | Population served | Percentage/% |
| Level 1 | 72,900 | 0.93 | 600,000 | 7.64 |
| Level 2 | 112,800 | 1.44 | 1,821,600 | 23.18 |
| Level 3 | 253,200 | 3.22 | 2,202,900 | 28.03 |
| Total | 438,900 | 5.59 | 4,084,500 | 58.85 |

sports parks. Therefore, optimization of the spatial distribution of sports parks in the following section was executed by improving the ratio of population with accessibility to sports parks.

### 3.3 Optimization of spatial distribution of sports parks in the study area

**3.3.1 Optimization strategies.** Spatial syntax theory is an important reference for optimizing the spatial distribution of sports parks in this study [16,28]. In this study, an axial analysis of the spatial structure of the study area was executed by using depth map to evaluate the integration and connectivity of the roads in the study area. Meanwhile, the results are visualized by using the ArcGIS (Figs 3 and 4). The colors of roads reflect the integration level (connectivity): red roads have the highest integration level (connectivity), while dark blue roads have the lowest integration level (connectivity). A high road integration indicates a high connectivity, therefore high utilization efficiency [31]. In other words, residents prefer a road with high integration level. Therefore, sports parks shall be arranged along high integrated/connected roads to maximize their accessibility [32].

As public facilities, sports parks should provide efficient and fair services. Due to the limited urban land, the number of sports parks should be rational and they should be located in densely populated areas. Specifically, the core blocks of the study area were identified based on the analysis of road network (Fig 3), and roads with low connectivity in the study area were excluded on the basis of global integration and global connectivity [33]. As observed, the area with the highest road density (the eastern part of Yuelu District, the northern part of Tianxin District, the northern part of Yuhua District, the western part of Furong District, and the southern part of Kaifu District) is essentially the central urban area of Changsha (i.e., the study area).

The optimization of spatial distribution of sports parks in the study area focused on maximization of population with accessibility to such a park, with maximization of area with accessibility to sports parks also taken into consideration.

**3.3.2 Optimization results.** According to the *Guiding Opinions on Promoting the Construction of Sports Parks* issued by National Development and Reform Commission of China, one sports park can serve up to 450,000 residents [34]. As the current population in the study area is 7.85 million, a total of 17 sports parks are needed, indicating that four new sports parks shall be constructed. Then, the spatial locations of new sports parks in the study area were determined. As mentioned above, sports parks shall be arranged along the roads with high integration and connectivity to maximize their accessibility. Herein, the spatial locations of new sports parks in the study area were determined based on the road network analysis shown in Fig 3. However, global integration and connectivity do not provide sufficient information for determination of new park locations. Hence, local integration (R3 and R5, see Fig 4) of the road network was analyzed, wherein R3 and R5 refers to three and five topological distances, respectively [35]. Specifically, roads with maximum local integration were identified and the new sports parks are located along these roads.

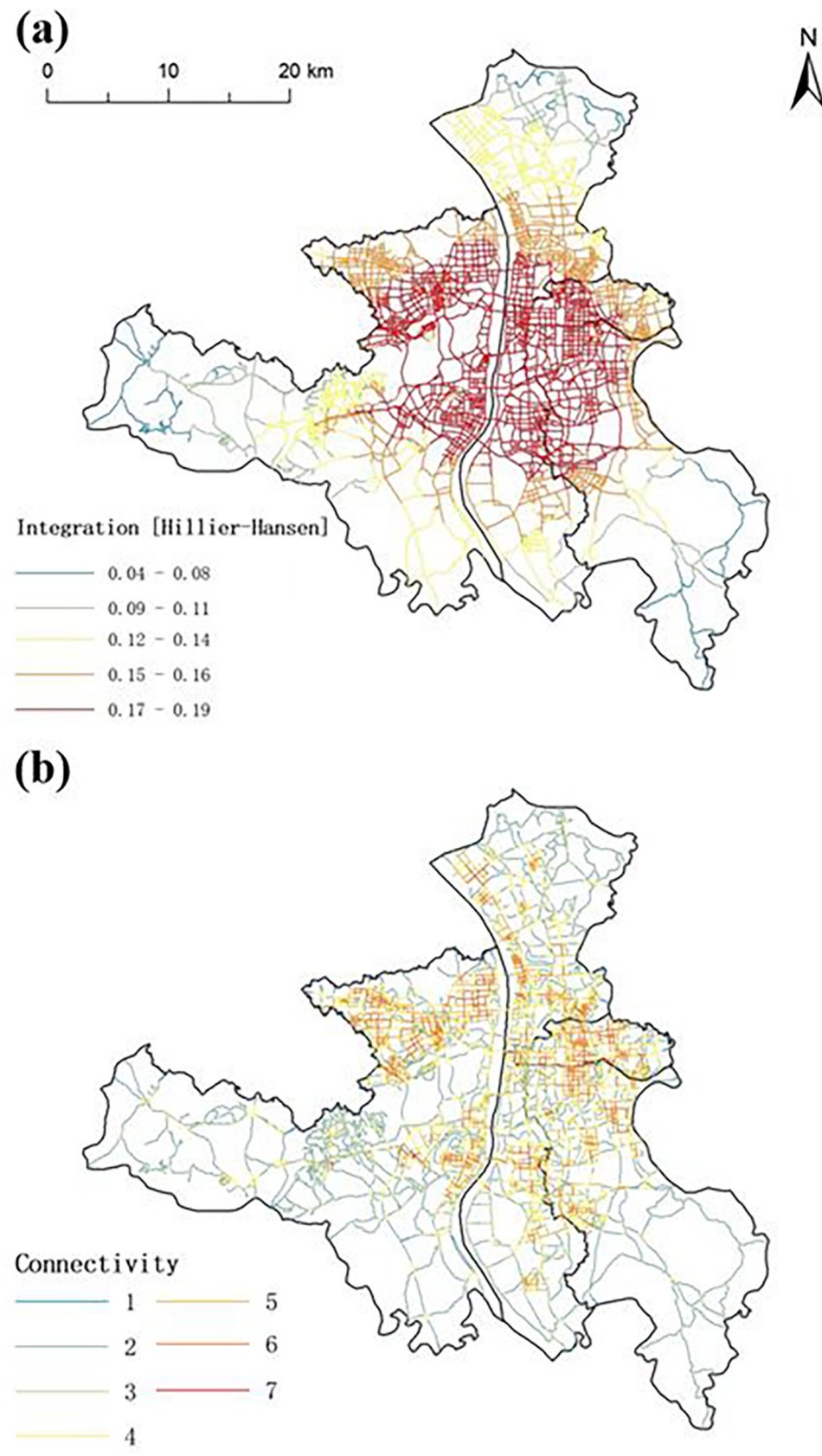

**Fig 3.** Analysis of road network in the study area in terms of (a) integration and (b) connectivity (Reprinted from Standard Map Service (http://bzdt.ch.mnr.gov.cn/) under a CC BY license, with permission from the Ministry of Natural Resources of the People's Republic of China, original copyright 2023).

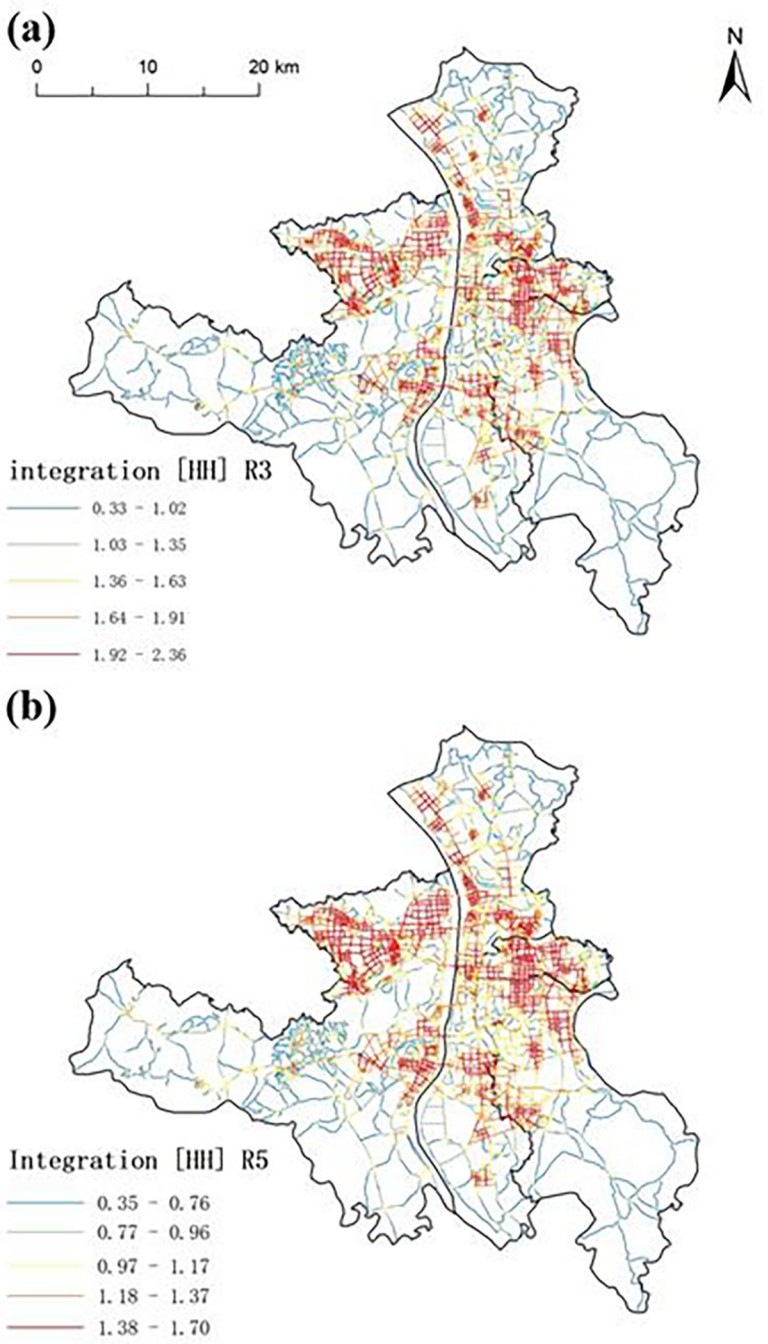

**Fig 4.** Local integration of road network in the study area: (a) three topological distances; (b) five topological distances (Reprinted from Standard Map Service (http://bzdt.ch.mnr.gov.cn/) under a CC BY license, with permission from the Ministry of Natural Resources of the People's Republic of China, original copyright 2023).

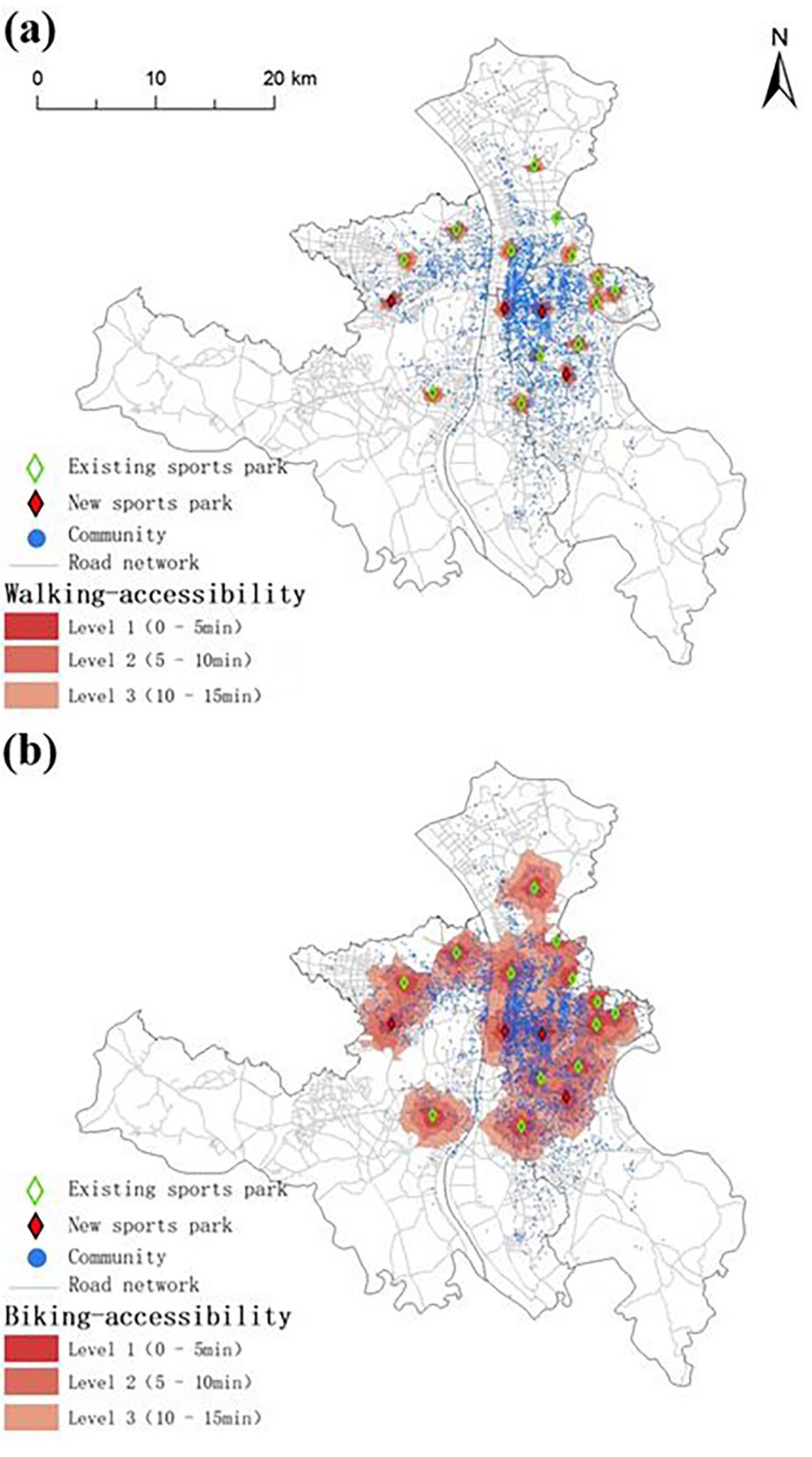

**Fig 5.** (a) Walking-accessibility and (b) biking-accessibility of existing and new sports parks in the study area (Reprinted from Standard Map Service (http://bzdt.ch.mnr.gov.cn/) under a CC BY license, with permission from the Ministry of Natural Resources of the People's Republic of China, original copyright 2023).

Fig 5 shows the proposed spatial locations of new sports parks, as well as walking- and biking-accessibility of existing and new sports parks in the study area evaluated using the network analysis method. As observed, both the area and population with accessibility to sports parks increased significantly after the development of new sports parks.

Table 6 summarizes the walking- and biking-accessibility of sports parks before and after optimization.

In terms of area with accessibility to sports parks, the cumulative area with walking-accessibility to sports parks in the study area increases from 2.08% to 2.91%, while the cumulative area with biking-accessibility to sports parks in the study area increases from 18.68% to 20.15%, suggesting that the optimization strategy has relatively low impacts on the area with biking-accessibility. Meanwhile, the area with Level 1 and 2 biking-accessibilities to sports parks in the study area increases from 2.55% to 3.47% and from 7.26% to 9.68%, respectively, after optimization, while the area with Level 3 biking-accessibility to sports parks in the study area decreases from 8.87% to 7.0%. This can be attributed to the increasing area with Level 1 and 2 biking-accessibilities, which coincides with the area with Level 3 biking-accessibility. Such changes indicate a favorable change in the structure of areas with biking-accessibility as most residents prefer level 1 and 2 accessibilities. In summary, the optimization is rational and effective as the area with Level 1 and 2 biking-accessibilities increases drastically, despite that the total increase of area with biking-accessibility induced by optimization is not significantly large.

In terms of population with accessibility to sports parks, the cumulative ratio of population with walking-accessibility to sports parks in the study area increases from 5.59% to 9.86%, and the ratios of population with Levels 1, 2 and 3 walking-accessibilities increase 52.69%, 86.11%, and 78.89%, respectively, indicating significant optimization of walking-accessibility of sports parks in the study area. Meanwhile, the cumulative ratio of population with biking-accessibility is 73.82%, and the ratio of population with Levels 1 and 2 biking-accessibilities is 46.12%. In other words, over 70% of the residents can reach a sports park by 15-minute biking, and approximately 50% of the residents can reach a sports park by 10-minute biking. In summary, the optimization is rational and effective.

After optimization (construction of four new sports parks), both area and population with walking-accessibility, as well as population with biking-accessibility to sports parks in the study area are significantly improved, despite that the area with biking-accessibility exhibits

**Table 6. Accessibility levels of sports parks in the study area before and after optimization.**

| Walking | Accessibility level | Before optimization | | After optimization | |
|---|---|---|---|---|---|
| | | Area ratio/% | Population ratio/% | Area ratio /% | Population ratio/% |
| | Level 1 | 0.22 | 0.93 | 0.29 | 1.42 |
| | Level 2 | 0.69 | 1.44 | 0.96 | 2.68 |
| | Level 3 | 1.17 | 3.22 | 1.66 | 5.76 |
| | In total | 2.08 | 5.59 | 2.91 | 9.86 |
| Biking | Level 1 | 2.55 | 7.64 | 3.47 | 12.27 |
| | Level 2 | 7.26 | 23.18 | 9.68 | 33.85 |
| | Level 3 | 8.87 | 28.03 | 7.00 | 27.70 |
| | In total | 18.68 | 58.85 | 20.15 | 73.82 |

negligible changes. This can be attributed to the concentration of residents in the core blocks of the study area. In summary, the locations of new sports parks are rational and the proposed optimization strategy is effective.

## 5 Conclusions and recommendations

The distribution of existing sports parks is not rational owing to the uneven population distribution. Specifically, the quantities of existing sports parks in different districts are not consistent with the populations of these districts. The walking-accessibility of existing sports parks in the study area is low and the biking-accessibility of existing sports parks in the study area is moderate, but can be further improved. Additionally, an optimization strategy for the spatial distribution of sports parks was proposed and the results demonstrated that construction of four new sports parks can lead to a significantly improved biking-accessibility to sports parks (78.89%).

This study provides a thorough investigation of walking- and biking-accessibility of sports parks in the study area, as well as a rational and effective optimization strategy for the. Nevertheless, this study has several limitations. Firstly, the characteristics of each sports park, including size, facilities and fee (if any), have not been taken into consideration. Secondly, the availability of public bikes, which may have impacts on the biking-accessibility to sports parks, have not been taken into consideration.

## Supporting information

**S1 File. Questionnaire.**
(DOCX)

## Author Contributions

**Conceptualization:** Hengtao Guo.

**Methodology:** Kairan Yang, Yujun Xie.

**Writing – original draft:** Kairan Yang.

**Writing – review & editing:** Hengtao Guo.

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
